# Fungal Endophytes for Grass Based Bioremediation: An Endophytic Consortium Isolated from *Agrostis stolonifera* Stimulates the Growth of *Festuca arundinacea* in Lead Contaminated Soil

**DOI:** 10.3390/jof6040254

**Published:** 2020-10-29

**Authors:** Erika Soldi, Catelyn Casey, Brian R. Murphy, Trevor R. Hodkinson

**Affiliations:** School of Natural Sciences and Trinity Centre for Biodiversity Research, Trinity College Dublin, College Green, The University of Dublin, D02 PN40 Dublin 2, Ireland; caseycj95@gmail.com (C.C.); murphb16@tcd.ie (B.R.M.); hodkinst@tcd.ie (T.R.H.)

**Keywords:** *Agrostis stolonifera*, bioremediation, endophytes, *Festuca arundinacea*, fungi, lead-contaminated soils

## Abstract

Bioremediation is an ecologically-friendly approach for the restoration of heavy metal-contaminated sites and can exploit environmental microorganisms such as bacteria and fungi. These microorganisms are capable of removing and/or deactivating pollutants from contaminated substrates through biological and chemical reactions. Moreover, they interact with the natural flora, protecting and stimulating plant growth in these harsh conditions. In this study, we isolated a group of endophytic fungi from *Agrostis stolonifera* grasses growing on toxic waste from an abandoned lead mine (up to 47,990 Pb mg/kg) and identified them using DNA sequencing (nrITS barcoding). The endophytes were then tested as a consortium of eight strains in a growth chamber experiment in association with the grass *Festuca arundinacea* at increasing concentrations of lead in the soil to investigate how they influenced several growth parameters. As a general trend, plants treated with endophytes performed better compared to the controls at each concentration of heavy metal, with significant improvements in growth recorded at the highest concentration of lead (800 galena mg/kg). Indeed, this set of plants germinated and tillered significantly earlier compared to the control, with greater production of foliar fresh and dry biomass. Compared with the control, endophyte treated plants germinated more than 1-day earlier and produced 35.91% more plant tillers at 35 days-after-sowing. Our results demonstrate the potential of these fungal endophytes used in a consortium for establishing grassy plant species on lead contaminated soils, which may result in practical applications for heavy metal bioremediation.

## 1. Introduction

Lead (Pb) is classified as a “heavy metal” due to its high relative atomic weight (>4 g/cm^3^). Some metals are essential to plant growth such as zinc and manganese, but lead does not have any known biological function and is toxic to plants even at low concentration [1]. It is, therefore, known as a toxic heavy metal (THM) and causes high levels of environmental pollution due to its extreme toxicity [2,3]. Indeed, lead is naturally present in rock formations within the Earth’s crust that becomes increasingly available in the biosphere due to human interference. Mining, ore processing, and Pb-acid battery recycling are the main sources, followed by agricultural practices (pesticides) and industrial products (dyes, paints, pipes) [1,2]. Once exposed on the Earth’s surface, the metal can be found in the atmosphere, hydrosphere, and pedosphere bound with other compounds. However, due to its non-biodegradable nature, it persists in the environment for a long time and becomes available for absorption and accumulation by microorganisms and plants, entering into the food chain and consequently negatively affecting animal and human health [2].

Due to the high toxicity of lead, plants activate several responses in the presence of the metal including (1) immobilization of the metal in the rhizosphere through the release of exudates or biosorption in the roots walls (exclusion); (2) assimilation of the metal into the root and activation of detoxification reactions involving the sequestration of the metal through the production of metal ligands such as phytochelatin and metallothioneins, and sometimes accumulation in cellular vacuoles (detoxification); and (3) activation of the NO-dependent stress response and other metabolic pathways to minimize the amount of bioavailable lead (Pb^2+^) in the cells (non-specific defense mechanism) [4,5]. Moreover, reactive oxygen species (ROS) are produced to facilitate the scavenging of the metal and to induce structural changes in the root’s cell walls to bind the toxic element [2,4]. However, the majority of the metal remains localized at the root level due to the Casparian strip of the endodermis, which acts as a barrier [5]. Once these defenses are overcome by high concentrations of lead in plants, the metal can damage the plant in many ways including germination abortion, reduction of root and shoot growth, photosynthesis inhibition, interruption of biological reactions where the metal acts as a “molecular mimic”, and a decline of anti-oxidant enzymes with a consequent increment of ROS and oxidative cellular damage [2,5,6]. 

Lead causes huge environmental and human health problems [7,8]. Therefore, detoxification of lead polluted sites is a major challenge. Bioremediation is an innovative and environmentally-friendly approach that can use microorganisms for the decontamination of substrates polluted by heavy metals. Indeed, many microbes are well-adapted to extreme environmental conditions, having several mechanisms of detoxification [9,10,11,12]. Microorganisms can reduce the amount of toxic metal in the soil by absorbing it into their membranes (biosorption), changing it chemically into a more innocuous form and positively stimulating their absorption by the plant (phytoremediation), helping to build up the organic matter in the soil [9,10,11,12].

Endophytes offer bioremediation potential [13]. Endophytes are microorganisms such as bacteria or fungi that live within plant tissue without causing any apparent disease and sometimes form symbiotic/mutualistic relationships [14]. These endo-organisms help the plant to better cope with abiotic and biotic stresses stimulating their growth and reducing the effects of stress on the plant [15].

Endophytes are also able to support plant growth in the presence of toxic heavy metals positively influencing the plants’ absorption, degradation, and dislocation of the contaminant [13]. For example, *Neotyphodium* are foliar endophytes that have been studied for their beneficial effect on grasses such as *Festuca* spp. and *Lolium* spp. in soil polluted with aluminum, zinc, copper, and cadmium [16,17,18,19]. Soleimani et al. [20] demonstrated that *Festuca* sp. treated with *N. coenophialum* and *N. uncinatum* produced a greater foliar and root biomass compared with untreated plants at a high concentration of cadmium in a hydroponic system. 

Recent research has emphasized the importance of using endophytes as a consortium rather than as a single strain. Indeed, Wężowicz et al. [21] found an incremental biomass production of *Verbascum lychnitis* grown in post-mining waste when co-inoculated with *Rhizopagus irregularis*, an arbuscular mycorrhizal fungus (AMF), *Cochliobolus sativus*, *Diaporthe* sp., and *Phoma exigua* var. *exigua* compared with non-inoculated and AMF plants. When the plants were inoculated only with *Diaporthe* sp., a negative effect was measured in the plant, an effect that was offset with a positive response when the plant was also inoculated with the AMF [21]. Vandegrift et al. [22], in a similar experiment, observed a reduction in the negative effect that a dark septate endophyte (DSE) had on *Agrostis capillaris* when co-inoculated with an *Epichloë* species. They hypothesized that excessive production of nitrogen by the DSE was used for the production of alkaloids for plant defense by the *Epichloë* species, balancing the costs for the plant in hosting the DSE [22]. 

However, relatively little is known about fungal endophytes and their potential application in bioremediation [13]. Therefore, to address this gap, in this study, a consortium of fungal root endophytes was isolated from grass plants (*Agrostis stolonifera*) from soil highly contaminated with lead. The aim of the research was first to establish the identity of the endophytes recovered from the grass and to investigate the literature for any bioremediation effects reported for these microorganisms. Once this was established, and considering their coexistence in the same sampled plant, the microorganisms selected from the recovered endophytes were tested as a consortium in a laboratory experiment in association with *Festuca arundinacea* Schreb. (also known as *Lolium arundinaceum* (Schreb.) Darbysh). The grass was grown in the presence of increasing concentrations of lead in the soil to determine the influence of the microbial consortium on plant metal toxicity.

## 2. Materials and Methods 

### 2.1. Study Site

Plant samples were collected from an abandoned mine site in Ballycorus, County Dublin, Ireland. The mine was manually excavated to recover lead and silver in the 20th century up until the 1920s. Previous research revealed an average lead concentration at the site of 1.5% (1.5% equal to 1500 Pb mg/kg) in spoil and slag waste, which exceeds the UK guideline limit of lead for soils (soil guideline values = SGV) on industrial sites (> 750 mg/kg Pb). The site (Figure 2) includes three solid waste heaps (designated SP01, SP02, and SP04) and an additional waste heap on the banks of the Loughlinstown River (SLAG01) [23]. 

SP02 is the largest and highest of the three heaps closest to the top of the mine, which constitutes the main volume of solid waste. The total area is 1505 m^2^ and its altitude ranges between 205 to 225 m along a northwest 10–30° slope. The soil is dry and loose, and vegetation is limited to sparsely distributed grasses such as *Agrostis stolonifera* and the site is surrounded by birch trees (*Betula* sp.) [24]. Concentrations of lead measured in this slope region were the highest with values ranging between 13,057 and 47,990 Pb mg/kg [23]. A recent analysis by Jordan [25] reported that the concentrations of lead at the edge, where birch trees were growing, had similar values to the previous measurements. Indeed on the northeast side of the slope, the lead measured between 38,911.5 to 20,557 Pb mg/kg, while on the northwest side, the concentration ranged between 16.787 to 11,219.5 Pb mg/kg [25]. SP01 is the middle heap with an area of 365 m^2^ and altitude between 188 to 194 m and features a northwest facing 10–20° slope. It is populated by *Agrostis stolonifera*, *Festuca ovina*, and *Galium verum*. Gorse (*Ulex europaeus*), pine (*Pinus sylvestris*), birch (*Betula* sp.), and bracken (*Pteridium aquifolium*) are also found in the surrounding area. The soil is equally as dry and rocky as SP02, but with no humic or mossy layer [24]. Lead in this heap was less concentrated compared to SP02, ranging between 6561 and 13,057 Pb mg/kg [23]. SP04 is the smallest spoil heap with an area of 148 m^2^. The heap is between 177 and 183 meters above sea level and has a northwest slope of between 10–30°. The area is surrounded by various plant species including *Rubus fruticosus* and *Ulex europaeus*, with a surface layer of moss and loose, rocky soil [24]. Lead concentrations at this site are unknown. 

Other minerals such as silver, zinc, nickel, and barium were found throughout the mine site at varying concentrations (see [23]).

### 2.2. Plant and Soil Sampling 

Plants for endophyte isolation were collected from the three waste heaps described above. Two *Agrostis stolonifera* plants were sampled at each of the three spoil heaps (SP01, SP02, and SP04) using a small spade without distressing the plant and root system. Isolated grass clumps were chosen away from the main grass carpet to avoid areas where the soil was restored and was therefore less rich in heavy metals. 

Once in the laboratory, five root samples were collected from each plant and surface sterilized by first washing each root sample in 70% ethanol and then submerging in 5% sodium hypochlorite for 3–5 min. Then, the roots were washed three times in deionized sterile water to remove any chemical residues. Each of the roots was cut into five non-contiguous segments of 1 cm length, which were then laid on ½ strength potato dextrose agar (PDA, VWR Chemicals) in Petri dishes. The plates were labelled, sealed with parafilm and stored in the dark at 20 °C in an incubator. The Petri dishes were observed daily until the endophytes began to grow out from either end of the root about a week after the culturing. Each emergent endophyte was isolated on a separate Petri dish and a minimum of three subcultures were established on separate plates.

### 2.3. DNA Analysis 

#### 2.3.1. DNA Extraction

For the identification of the endophytes, total genomic DNA was isolated using a CTAB (Hexadecyltrimethylammonium bromide) extraction in 2 mL microcentrifuge tubes as follows: the extraction buffer was prepared by adding 1 mL of 2× CTAB solution and 0.2% β-mercaptoethanol into a sterile 2 mL microcentrifuge tube and was preheated in a heat block at 65 °C for 15 min. Fungal material of 200 mg average weight for each sample was added to the preheated buffer with a metal bead and the contents were homogenized using a Mixer Mill MM 300 (Retsch) for 1.5 minutes at 30 Hz each side. The samples were then incubated at 65 °C for one hour, mixing the tubes by inversion 3–4 times. A volume of 500 µL of chloroform:isoamylalcohol (24:1) was added to each tube, which were shaken vigorously for 15 seconds and laid on a shaking plate at 220 rpm for 10 min to mix well. The samples were centrifuged at 8000× *g* for 10 min and the upper CTAB phase of approximately 900 µL was transferred to a new sterile 1.5 mL tube. An equal volume of 500 µL ice-cold isopropanol was added to each tube, which were mixed a few times by inversion and stored at −20 °C for an hour. The tubes were centrifuged at 8000× *g* for 5 min to spin down the DNA. The supernatant was removed, and the pellet was washed with 800 µL of 70% ethanol and the samples were centrifuged at 8000× *g* for 3 min. The supernatant was removed, and the samples were left to dry at room temperature under a chemical hood for between 30 minutes to one hour. The pellets were resuspended in 100 µL TE buffer and the samples were kept at 37 °C in a heat box for 30 min to allow the DNA to fully dissolve. The final concentrations of DNA (ng·µL^−1^) in the extracted samples was measured with a Nanodrop UV spectrophotometer (NanoDrop Lite, Thermo Scientific, Waltham, MA, USA). 

#### 2.3.2. Polymerase Chain Reaction (PCR) Cycle, Sequencing and Identification

The fungal isolates were grouped based on their colony morphology as observed on the ½ strength potato dextrose agar (PDA). Twelve different morphologies were observed, out of which eight were chosen to use for further analysis based on their rapid growth at room temperature and their early sporulation.

Amplification of the nrITS barcode intergenic regions were carried out using the primers ITS1F (5′-CTTGGTCATTTAGAGGAAGTAA-3′) [26] and ITS4 (5′- TCCTCCGCTTATTGATATGC-3′) [27] using the following PCR cycling parameters: initial denaturation at 94 °C for 3 min; followed by 38 cycles of denaturation at 94 °C for 1 min, annealing at 55 °C for 1 min, extension at 72 °C for 1 min; a final extension phase for 10 min at 72 °C. PCR products were cleaned up using the ExoSAP-IT™ PCR Product Cleanup Reagent (Thermo Fisher Scientific) and final products were sequenced using the Sanger Sequencing Service offered by Source Biosciences. Nucleotide sequences were identified by comparing them against the NCBI (National Center for Biotechnology Information, Bethesda, MD, USA) GenBank and the UNITE database (Unified system for the DNA based fungal species linked to the classification, https://unite.ut.ee). Similarity criteria for assigning taxonomic rank to the endophyte strains was allocated based on an initial survey of existing fungal taxa in UNITE and GenBank, as follows: >97% similarity was assigned to the same species, 90–96% to the same genus, 85–90% to the same order, and <85% to no significant match.

Furthermore, β–tubulin (Bta2) and translation elongation factor 1-α (tef) sequences were amplified following the cycle described by Raja et al. (2017) to confirm the identification of some samples [28].

### 2.4. Growth Experiment In Vivo of Festuca arundinacea with Lead

#### 2.4.1. Endophyte Inoculum

For the inoculum, single endophyte isolate spore solutions were prepared by washing the fungal colony with sterile water and delicately shaking the plate to dislodge the spores into the solution that was then collected. The concentration of spores was measured using a hemocytometer and the solution modified to give a concentration of 10^6^ spores/mL. Each isolate spore solution (eight in total) were mixed in equal quantities to reach a final inoculum volume of 12 mL.

#### 2.4.2. Plant and Pots Preparation 

Lead is naturally bound with other elements and is commonly found as galena. Galena (mainly PbS), containing on average 86% of Pb, 13% of S, and 1% Ag is the main ore of lead [29]. This mineral can also be found in the Ballycorus mine site, therefore, it was used for our purpose as a source of lead to mimic the original environment from where the endophytes were isolated. 

In the following experiment, four incremental concentrations of galena (Pb_0_ = 0, Pb_400_ = 400, Pb_600_ = 600, Pb_800_ = 800 galena mg/kg) were mixed separately with 5 kg of John Innes No. 2 compost (Westland Garden Health, pH 5.5–6.0 at 20 °C, https://www.gardenhealth.com/westland-john-innes-no-2-potting-on-compost) and then, divided between 20 pots of 1 L volume for a total of 80 pots. Then, two seeds of *Festuca arundinacea* (cultivar Jesup, endophytes-free Nil, type continental, accession number: T9904, % seedling emergence of 92%; supplied by Grasslanz Technology Limited, New Zealand) were sown in each pot at a 1 cm depth. The 20 pots for each concentration of galena were equally split between the control and endophytic set by inoculating the seeds with either 200 µL of sterile water for the control or 200 µL of the microbial inoculate for the endophytes set, for a total of 10 pots for each treatment set (see Figure 1). The pots were then labelled and randomly distributed in the growth cabinet. 

The growth cabinet (Conviron PGR14) was programmed to produce a 16 h photoperiod at a compost surface illumination of 230 µmol m^−2^ s^−1^, a photoperiod temperature of 20 °C, reducing to 10 °C in the dark period and a constant 65% relative humidity. 

The plants were given no supplementary nutrients during the experimental period and were harvested after 50 days. Germination and tillering time were recorded for each plant during the course of the experiment and fresh and dry weight (g) of above ground tissue was measured post-harvest. Foliar biomass from plants in the same pot were collected and measured as a unique sample to assess the fresh foliar weight per pot, which were then dried in the oven at 65 °C for three days before measuring the dry weight. 

### 2.5. Data Analysis during the Experiment and Post-Harvest

#### 2.5.1. Germination and Tillering

The germination index (GI), described by formula (1), was designed to qualitatively compare the seed germination rate and the timing of germination between each of the treatment sets. A higher GI index means a higher number of seeds germinated at earlier times versus lower index, which described a set with a lower number of seed germinated. Indeed, for each seed that did not germinate, the index value was heavily penalized.

(1) GI = (N seed germinated ≤ 15d) × 4 + (N seed germinated > 15d) × 2 − (N seed NOT germinated) × 5 

The GI was calculated by summing two factors: the number of seeds that germinated before 15 days-after-sowing (das) were multiplied by 4 and this figure was added to the number of seeds that germinated after 15 days multiplied by 2. Indeed, 92.50% of seeds germinated before 15 das (days-after-sowing). The number of seeds that did not germinate was multiplied by 5 and subtracted from the total. 

#### 2.5.2. Post-Harvest Data and Statistical Analysis 

Boxplots were drawn using Microsoft Excel 365© to statistically assess germination and tillering, and for comparing post-harvest plant biomass. Boxplots were chosen as they visually show the distribution of numerical data and skewness through displaying the data quartiles (or percentiles) and averages. Data analysis was further carried out using single factor and two-way analysis of variance (ANOVA) in Microsoft Excel 365© and IBM SPSS Statistics (V27, Dublin, IR).

## 3. Results

### 3.1. Fungal Endophytes Isolated from Sampled Plants

A total of 62 microorganism cultures were isolated from *Agrostis stolonifera* roots collected from the three waste heaps at the Ballycorus mine site (Figure 2).

From among these endophyte cultures, eight were selected and identified using nrITS DNA sequences and they have been deposited in GenBank (Table 1). Additionally, β-tubulin (Bta) and translation elongation factor 1-α (tef) sequences were used to confirm the identifications (data not shown). Sequences of six microorganisms showed pairwise similarities of over 98% when compared using BLAST to the sequences in GenBank and UNITE and were identified to the species-level. The isolates TCDAs18R2A9 showed pairwise similarities of between 93 and 98% compared with the reference query, so it was identified to genus-level. 

### 3.2. In Vivo Tests with Festuca arundinacea, Endophytes, and Contrasting Levels of Lead in the Soil 

#### 3.2.1. Germination and Tillering

A higher concentration of lead in the soils negatively influenced the germination of *Festuca arundinacea* seeds as revealed by two-way ANOVA analysis. Indeed, increasing heavy metal concentration in the soil further delayed germination with statistical significance for each set (Pb_0_, Pb_400_, Pb_600_, Pb_800_) (*p* < 0.01). However, overall, the two-way ANOVA analysis also demonstrated that *Festuca arundinacea* seeds treated with endophytes (*E*) germinated significantly earlier than the control set (*p* < 0.05) (Figure 3). In particular, the germination of endophyte treated plants at the highest concentration of lead in the soils (*E* Pb_800_) was significantly earlier compared with the control (*C* Pb_800_) (*p* < 0.05).

Moreover, no significant results were obtained when the interaction between endophyte and control treatments (E vs. C) and concentration of lead in the soil (Pb_0_ Pb_400_, Pb_600_, and Pb800) were analyzed. This result confirmed that the effects induced by the endophytes were not influenced by increasing concentration of lead in the soil.

Furthermore, a greater germination index (GI) was measured for each set of endophyte treatments at different concentrations of lead in the soil, which means that a greater number of seeds sprouted compared with the untreated sets (see below, Table 2). Additionally, a significantly greater number of plants treated with endophytes at 600 galena mg/kg (*E* Pb_600_) germinated compared with the control set (*C* Pb_600_) (*p* < 0.05). 

Generally, plants treated with endophytes tillered earlier compared with the control treatments (two-way ANOVA, *p* < 0.05). Indeed, more than 90% of endophyte treated plants were tillering at 35 days-after-sowing with an average of 13.08% more tillered plants compared with the control. Moreover, plants at treatment E Pb_800_ had higher percentage tillering than untreated plants at 35 days-after-sowing (*p* < 0.01) (Figure 4).

#### 3.2.2. Post-Harvest Analysis

A higher concentration of lead in the soils negatively influenced the biomass production by *Festuca arundinacea* as revealed by two-way ANOVA. Increasing heavy metal concentration in the soil induced a significant reduction of the fresh and dry biomass for each set (Pb_0_ Pb_400_, Pb_600_, Pb_800_) (*p* < 0.01).

Due to the unequal number of germinated plants between treatments, it was not possible to compare the total biomass produced by each treatment set. However, no major differences were observed in the average foliar biomass between untreated plants (*C*) and plants treated with endophytes (*E*) (Figure 5a,b) with the exception of plants grown in soil with the highest concentration of lead (*E* Pb_800_). This set of plants produced a greater fresh and dry foliar biomass compared with the control set. However, no statistical significance was observed due the high variability within the dataset, here represented by the box size and the bars in the box plots (Figure 5a,b).

The production of tillers was also generally negatively affected by an increasing concentration of lead in the soil (Figure 6). However, as a general trend, plants treated with endophytes produced on average more tillers per pot compared with the control set, with major differences observed between the sets grown at the highest concentration of lead (Pb_800_).

## 4. Discussion

Little is still known about the role of fungal endophytes in supporting the growth of plants in soil contaminated with toxic heavy metals such as lead. Therefore, in this study, root endophytes were isolated from grasses that were growing in an abandoned lead mine. The endophytes were inoculated onto the grass *Festuca arundinacea* in a laboratory experiment to assess the impact of the endophyte consortium on this host at increasing concentrations of lead. Results demonstrated a better performance of the plants treated with the endophyte consortium for each of the growth parameters recorded, hence reducing the detrimental effect of the heavy metal. 

Grasses such as *Agrostis* spp. and *Festuca* spp. generally grow relatively well in heavy metal contaminated soils, even in the absence of any symbiotic microbiota, due to a local evolutionary adaptation leading to heavy metal tolerant ecotypes [30,31]. However, in the current study, this aspect was not considered. 

A total of 62 fungal cultures were isolated in the laboratory from the root tissue of *Agrostis stolonifera* (creeping bent grass); eight of which were selected and further identified. Other research has demonstrated that the abundance and diversity of the microbial community is negatively correlated with a high concentration of toxic heavy metal in the soil, although populated by more resistance strains [32,33]. Thus, the plants collected were growing away from the main grass carpet to increase the probability of obtaining a more heavy metal resistance organism. 

All the different genera isolated belonged to different fungal orders and three different classes of fungi were represented (Sordariomycetes: *Acremonium*, *Sarocladium* and *Tolypocladium*; Eurotiomycetes: *Penicillium*; Leotiomycetes: *Phialocephala*). Four isolates were *Penicillium* species and this was the dominant genus isolated (as identified by nrITS and colony morphology under microscope), which was not unexpected considering their high tolerance and efficiency in heavy metal detoxification [11]. *Tolypocladium* spp. and *Phialocephala* spp. have been isolated in previous studies in association with toxic heavy metals [33,34], whereas *Acremonium* spp. and *Sarocladium* spp. were for the first time connected with this topic. 

The eight endophytes described above were then used as a consortium in a laboratory experiment in association with *Festuca arundinacea* to test their role in plant growth and development in lead-contaminated soil. The organisms were recovered from the same environment and coexisted within the same plant species from which they were isolated (*Agrostis stolonifera*). Therefore, a beneficial effect was predicted in using these fungi as a unique inoculum rather than separately as single strains. Indeed, many previous studies observed a better performance of the endophytes when used as a consortium rather than single endophyte species [21,22,35,36]. For example, Ojuederie and Babalola [9] reported in their review a synergistic effect induced by four bacterial strains that improved the remediation of soil contaminated by lead, cadmium, and copper when they were used as a consortium [37]. Wężowicz et al. [21] measured a significant incremental production of chlorophyll *b* when *Verbascum lychnitis* was co-inoculated with *Xylaria* sp. and *Diaporthe* sp.; the effects were not visible if the plants were inoculated with a single species. Indeed, the biosynthetic pathway of this chlorophyll is particularly negatively influenced by Pb; therefore, the authors declared that the inoculation of a endophytic consortium positively impacts the composition of the photosynthesis system, which was consequently associated with an increase in photosynthesis efficiency for plants grown in heavy metal conditions [21]. 

Toxic effects of lead on the plants were observed during this experiment in each parameter measured, from the delay in germination to the decrease in harvested foliar biomass. Shama and Dubey [5] reported that the Pb negatively influenced the synthesis of biomolecules such as nucleic acids and proteins in seed and also altered plant tissue protein and lipid composition, interfering with biological processes in the plant.

However, it was observed in this study that *Festuca arundinacea* treated with endophytes overall performed better in the first stage of development of the plant with statistical significance compared with the control set, with particular improvements for plants grown at the highest concentration of lead (*E* Pb_800_). 

Endophytes have been found to positively influence seed germination and stimulate plant growth though the production of plant-like hormones under stress conditions [38]. Leitão et al. [39] described in detail the role of endophytes in producing plant-like gibberellins when they interacted with a host: they found that *Penicillium*-produced gibberellins not only positively stimulated the growth of the plant in conditions of high salinity, but also altered the abscisic acid (ABA), jasmonic acid (JA), and salicylic acid (SA) levels in the host plant and reduced ROS in the intracellular plant environment, alleviating the negative effect of the salt [39]. Similar mechanisms might have been employed by the isolates in the current study, stimulating the germination and development of *Festuca arundinacea*, but further experiments are required to confirm that.

Overall, a positive influence by the endophytes in mitigating the effect of the heavy metal in *F. arundinacea* plants was demonstrated here, with major effects at the highest concentration of lead (*E* Pb_800_ vs *C* Pb_800_). The mechanisms by which endophytes positively influenced the fitness of the plants were not the focus of this study. However, some assumptions can be made. 

Bibi et al. [40] demonstrated that *Lactuca sativa* grown in cadmium contaminated soil and treated with an endophytic consortium had greater foliar and root biomass compared with the untreated set. They also demonstrated that the application of the endophytes not only improved plant fitness in stressful conditions, but also reduced the accumulation of cadmium in the edible part of the lettuce, reducing the risk for human health [40]. 

Similar results were found by Li et al. [41], who inoculated maize with the fungal endophyte *Exophiala pisciphila* H93 in the presence of toxic heavy metals including lead, zinc, and cadmium, and they observed higher crop biomass production compared with the untreated set [41]. 

Both the above experiments highlighted a correlation between a reduction in the concentration of metal in the shoot and a better performance of the crops in the presence of the endophytes, which, they deduced, locked the metal in soils or roots, reducing its bioavailability in the plant system. Endophytes generally reduce the bioavailability of Pb^2+^ ions in the plant’s cells through mechanisms of metal exclusion and stabilization in the soil (phytostabilization) or metal extraction from the soil and compartmentalization in specific organs such as microbial membranes and plant vacuoles (phytoextraction) [3,9,11,13]. 

Three species of *Penicillium* were identified among the endophytes isolated in this study. *Penicillium* spp. have been extensively studied for their ability of biosorption and bioaccumulation of toxic heavy metals [11,13,42]. These fungal species are characterized by an agglomerate of polysaccharides and proteins, which are negatively charged and can effect cation exchange and immobilize metals [11]. In another study, one of the species isolated here, *P. canescens* (TCDAs18R2A2 and TCDAs18R2A7), was shown to strongly biosorb into its membranes toxic heavy metals, with largely higher affinity for lead (Pb^2+^) compared to cadmium (Cd^2+^), mercury (Hg^2+^), and arsenic (As^3+^) [43]. Furthermore, Herrera et al. [44] found that *P. sanguifluum* (TCDAs18R2A6) was capable of producing siderophores, which, in this particular condition, can be released in the soil and bind the heavy metal, thus reducing the absorption by the plant roots [45].

*Phialocephala* spp. (TCDAs18R1B3) are sometimes classified as dark septate endophytes (DSEs) [46], which are often found in association with plants growing in heavy metal conditions and can reduce phytotoxicity [41,47,48,49,50]. Unfortunately, the mechanisms involved in the interaction between plant and these root endophytes are not yet clear. However, these microorganisms use methods to reduce the bioavailability of the metal in their growing environment, which can include the plant cell environment, and consequently can reduce the damage induced by the metal in the host. An increase in membrane thickness and melanin concentration in the membrane and increasing production of anti-oxidant enzymes, glutathione ligands, and siderophores by DSEs were positively correlated with an increasing concentration of metal in the growth media, with consequent increased binding ability of the metal, reducing side effects [49,50,51].

*Tolypocladium cylindrosporum* (TCDAs18R1A9) has not previously been isolated on soil contaminated with heavy metal. However, a closely related species, *T. inflatum*, is commonly found in Norwegian soil with elevated concentrations of lead originating from galena deposits and was found to be capable of growing in high metal concentrations in media experiments (10 mM Pb) [33]. When the radial growth rate of these isolates in Pb-adjusted media was compared with strains isolated from Swedish non-contaminated soils, no differential behavior was observed. Indeed, all the isolates were well capable of growing at high concentrations of metal. Therefore, Bååth, Díaz-Raviña, and Bakken [33] declared an intrinsic tolerance of this species to lead. Furthermore, Gräfe et al. [47] isolated a new metabolite from *Tolypocladium inflatum*, Tolypocladin, which is capable of binding di- and trivalent metals. Unfortunately, lead was not included during this experiment; however, they demonstrated that Tolypocladin was capable of binding other heavy metals such as Zi^2+^, Cu^2+^, Co^2+^ and Ni^2+^, and causing their precipitation as complex salts in weak acid conditions [52]. 

*Acremonium* spp. (TCDAs18R2A9) are frequently associated with grasses such as tall fescue (*Festuca arundinacea*) and perennial ryegrass (*Lolium perenne*) and can beneficially interact with the host improving tolerance against biotic stresses such as herbivorous or insect attack, or abiotic stresses such as drought [53,54]. However, *Acremonium* spp. have not been isolated often from heavy metal contaminated soil except for isolates from the genus *Neotyphodium*, which were formerly part of the *Acremonium* section [55,56,57] and a few other examples. *Acremonium pinkertoniae* was isolated from iron smelter wastes in Villerupt (France) by Zapotoczny et al. [58], who observed morphological changes in the microorganism exposed to high concentrations of copper. They proposed that *A. pinkertoniae* was capable of growing at toxic levels of heavy metal using two mechanisms of detoxification: (1) avoidance, which involved the thickening of the membrane to limit the exposure of cells’ interiors, and (2) an adaptive tolerance, which involved increasing the chitin–glucan content that bound the metal in the wall membrane [58]. Mohammadian et al. [59] recurrently isolated *A. persicinum* from contaminated soil in a lead–zinc mine in Iran. The microorganism was able to grow in the presence of heavy metals and better biosorb Zn, Cd, and Pb compared with other fungal isolates [59].

Other than these detoxification mechanisms, fungal isolates are known to produce other kinds of compounds that could stimulate plant growth in stress conditions such as antioxidant enzymes, organic molecules (malic acid, etc.), plant-like hormones (IAA, etc.), and antimicrobial compounds [39,60,61,62,63,64].

To summarize, a combination of these detoxification microbial mechanisms may be involved in the improved performance of *F. arundinacea* in the presence of lead in our experiment. Regardless of the lack of knowledge on the actual mitigation mechanisms used by these microorganisms, fungal endophytes are very promising for the bioremediation and restoration of contaminated soils, not only for their detoxification effects, as demonstrated here, but also for their higher resistance to heavy metals than bacteria [13].

Future research should focus further on elucidating the mechanisms of heavy metal stress alleviation in the plant and on gaining new insights into the complex relationship between these fungi and their hosts. Thus, isolation of other endo-organisms from different plant tissue could be used to obtain a different variety of microorganisms, which could include, for example, *Epichloë* and AMF; this would allow us to identify the effect that different classes of microorganisms have in the plant. Indeed, Wężowicz et al. [21] and Vandegrift et al. [22] observed a better performance of the plants when organisms from different classes were combined. This approach will also further reduce the gap between a controlled environment and a natural environment, allowing us to recreate a more realistic natural scenario and microbiota.

## 5. Conclusions

Overall, these results confirmed the vital role of the endophytes for the survival and the quicker establishment of new vegetation in such extreme conditions of heavy metal stress, not only selectively interacting with just one plant such as *Agrostis* spp., but also showing a flexibility to interact with other species such as *Festuca*. Indeed, the quicker establishment of the plants in these conditions, induced by the endophytes, would accelerate the restoration of deteriorated areas, thus reducing the presence of bioactive metals and improving the ecology, variety of soil microflora, and soil fertility, and, where possible, reestablishing healthy crop production.

## Figures and Tables

**Figure 1 jof-06-00254-f001:**
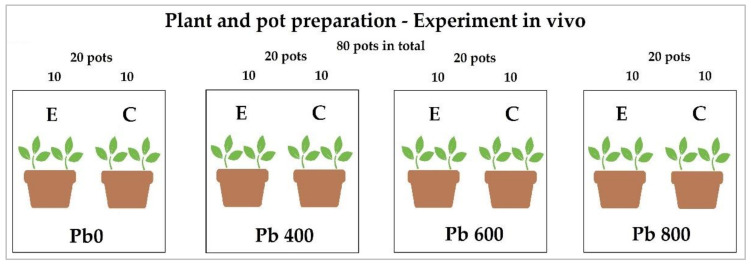
Design of the in vivo experiment. Galena at each concentration was added separately to the soil and well mixed before dividing into 20 pots for each treatment for a total of 80 pots. In each pot, two seeds of *Festuca arundinacea* were sown. Pots labelled *E* were treated with the endophyte consortium and those labeled *C* were treated with sterile water.

**Figure 2 jof-06-00254-f002:**
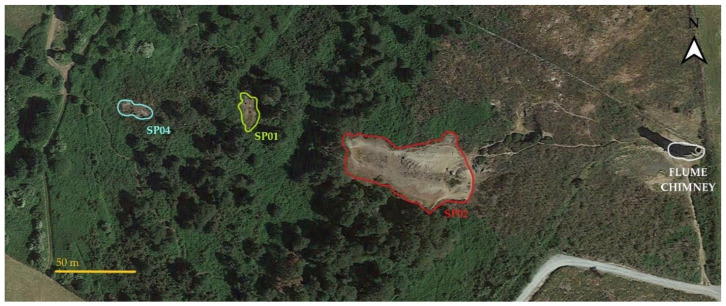
Ballycorus site map and spoil heaps (SP02, SP01, SP04). The flume chimney (highlighted in white) was the mine shaft site and was constructed in the early 1860s to vent the smelter smoke. SP02 was the original site used for lead ore and solid waste produced by these excavations are still accumulated at the edge of the heap representing the main volume of exposed mine waste. SP01 and SP04 are smaller solid waste heaps [23].

**Figure 3 jof-06-00254-f003:**
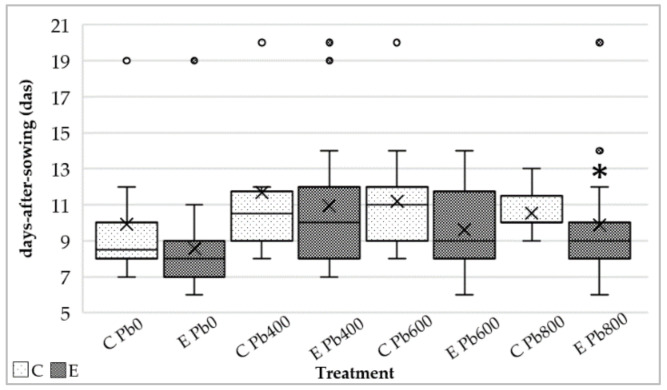
Germination time (here indicated as days-after-sowing, *das*) of *Festuca arundinacea* seeds. Endophytic treatment is indicated by the letter *E* and control by the letter *C* at increasing concentration of lead in the soil. The significant statistical difference (*p* < 0.05) between endophytic and control set is marked with a (*).

**Figure 4 jof-06-00254-f004:**
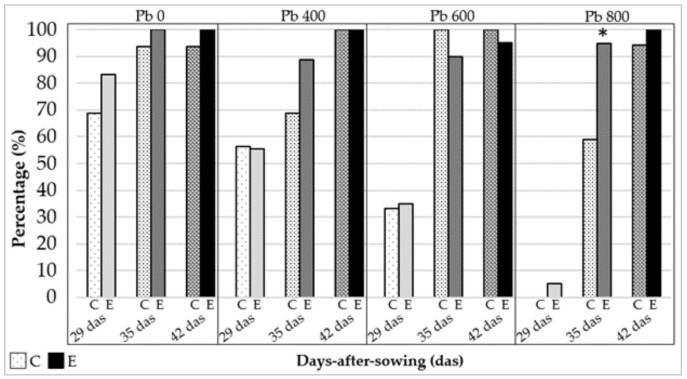
Percentage of total tillered plants at three recorded times (*29* das, *35* das, and *42* das) of *Festuca arundinacea*; endophytic treatment (*E*) and control (*C*) grown at increasing concentrations of lead in the soil. Significant statistical difference (*p* < 0.05) between endophytic and control set is marked with a (*).

**Figure 5 jof-06-00254-f005:**
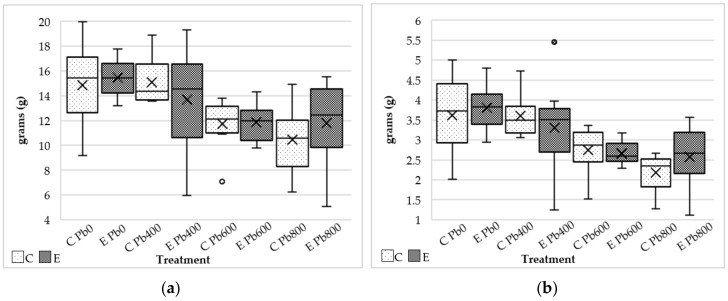
Foliar biomass measurement of fresh (**a**) and dry weight (**b**) of *Festuca arundinacea* with endophytes treatment (*E*) were compared with the control treatment (*C*) at increasing concentrations of lead in the soil.

**Figure 6 jof-06-00254-f006:**
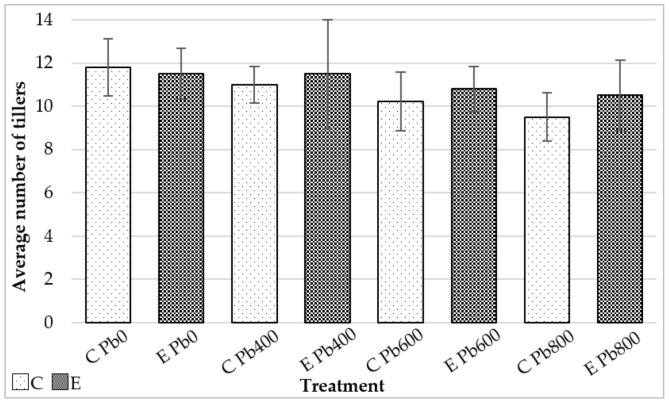
Average of total tillers per pot produced by *Festuca arundinacea* with endophyte treatment (*E*) were compared with control treatment (*C*) at increasing concentration of lead in the soil. Bars in the graph represent the standard deviation.

**Table 1 jof-06-00254-t001:** Identification summary of the experimental fungal endophyte strains. nrITS gene sequences of the isolates were compared using BLAST to those in the GenBank and UNITE databases.

Endophyte Strain ID	GenBank Accession	Nearest BLAST Match	Pairwise Similarity (%)	Query Reference
TCDAs18R1A3	MT911433	*Penicillium lanosum*	99.61	MH854599
TCDAs18R1A9	MT911434	*Tolypocladium cylindrosporum*	100.00	MH861264
TCDAs18R1B3	MT911435	*Phialocephala sp.*	99.82	JN995643
TCDAs18R2A2	MT911436	*Penicillium canescens*	100.00	MH865744
TCDAs18R2A6	MT911437	*Penicillium sanguifluum*	99.14	MH858377
TCDAs18R2A7	MT911438	*Penicillium canescens*	100.00	MH865744
TCDAs18R2A9	MT911439	*Acremonium sp.*	93.32	EU488735
TCDAs18R4B4	MT911440	*Sarocladium kiliense*	99.82	KM231849

**Table 2 jof-06-00254-t002:** Summary of germination index (GI) results. Number of germinated seeds of *Festuca arundinacea* with endophyte treatment (*E*) are compared with the control treatment (*C*) at increasing concentrations of lead in the soil. Significant statistical difference (*p* < 0.05) between endophytic and control set is marked with a (*).

Treatment	GI
Treatment C Pb_0_	40
Treatment E Pb_0_	65
Treatment C Pb_400_	38
Treatment E Pb_400_	54
Treatment C Pb_600_	33
Treatment E Pb_600_	75 (*)
Treatment C Pb_800_	53
Treatment E Pb_800_	69

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
