# Peer review of "Fungal Endophytes for Grass Based Bioremediation: An Endophytic Consortium Isolated from Agrostis stolonifera Stimulates the Growth of Festuca arundinacea in Lead Contaminated Soil"

_jof, 2020, doi:10.3390/jof6040254_

Round 1

Reviewer 1 Report

The authors isolated root endophytic fungi from Agrostis stolonifera at a lead contaminated region. The isolated fungi were identified based on ITS sequences. Festuca arundinacea were inoculated with endophytic fungi and their grow performance under lead levels were measured.

My main concern is on the endophyte inoculation strategy. Authors mixed spores from 8 fungal species in the suspension as inoculum. Comparing to test endophyte effects on host plant by single endophyte species, what’s the advantage of this strategy? Mixed inoculum, or “consortium” as used in title, would make interpretation to results difficult, and is not helpful to determine the function of each strain. I understand recent researches emphasized the importance of members in an endophyte community functioning as an entity. But if authors intended to investigate the endophyte effect at community level, related contents should be elaborated in the introduction. Actually, the term “consortium” has presented only once in the introduction, at the last paragraph. Thus I would suggest move some contents in Discussion section forward to Introduction and provide more information on this issue, in order to pave the path to your inoculation strategy. (Alternatively, the experiment design should include comparison between consortium and single species on the effects to plant performance, but I think it’s not necessary with proper statements in Introduction.)

In general, the manuscript is written in good structure and language, the results are clear and well represented. I would suggest acceptance after minor revision.

Comments in the text:

L122: “Other research …..” This sentence is not relevant here and should go to discussion.

L127: Five root samples from on plant. And how many pieces were incubated for one root sample?

L142 and 145: avoid starting a sentence with a number written in figures.

L184: Is it a double space behind “Ag,”?

L187: This part is not clear. Are there 4 treatments based on Pb concentration, and 10 pots for each treatment, which is 40 pots together? And 2 seeds in one pot makes the experiment included 80 replicates in 40 pots, is it correct? Then how do you inoculate spore suspension to the seeds? Are the non-inoculated and inoculated seeds in a same pots or in separate pots?

L216: Please mention briefly here how the fresh and dry biomass is measured.

L257: “(Pb0 = 0, Pb400 = 400, Pb600 = 600, Pb800 = 800 galena mg/kg)” is not needed to repeat in figure captions, since it has been indicated in method section.

L292: “(using box plots)” is not necessary.

Author Response

My main concern is on the endophyte inoculation strategy. Authors mixed spores from 8 fungal species in the suspension as inoculum. Comparing to test endophyte effects on host plant by single endophyte species, what’s the advantage of this strategy? Mixed inoculum, or “consortium” as used in title, would make interpretation to results difficult, and is not helpful to determine the function of each strain. I understand recent researches emphasized the importance of members in an endophyte community functioning as an entity. But if authors intended to investigate the endophyte effect at community level, related contents should be elaborated in the introduction. Actually, the term “consortium” has presented only once in the introduction, at the last paragraph. Thus I would suggest move some contents in Discussion section forward to Introduction and provide more information on this issue, in order to pave the path to your inoculation strategy. (Alternatively, the experiment design should include comparison between consortium and single species on the effects to plant performance, but I think it’s not necessary with proper statements in Introduction.)

Response and correction of the manuscript based on the comments (see document attached)

Recent research has emphasised the importance of using endophytes as a consortium rather than as a single strain. Indeed, Wężowicz et al. [21] found an incremental biomass production of Verbascum lychnitis grown in post-mining waste when co-inoculated with Rhizopagus irregularis an arbuscular mycorrhizal fungus (AMF) and Cochliobolus sativus, Diaporthe sp., and Phoma exigua var. exigua compared with non-inoculated and AMF plants. When the plants were inoculated only with Diaporthe sp. a negative effect was measured in the plant, an effect that was offset with a positive response when the plant was also inoculated with the AMF [21]. Vandegrift et al. [22], in a similar experiment observed a reduction of the negative effect that a dark septate endophyte (DSE) was having on Agrostis capillaris when co-inoculated with an Epichloë species. They hypothesized that excessive production of nitrogen by the DSE was used for the production of alkaloids for plant defence by the Epichloë species, balancing the costs for the plant in hosting the DSE [22].

Comments in the text: 

L122: “Other research …..” This sentence is not relevant here and should go to discussion.

Correction of the manuscript based on the comments (see document attached)

L127: Five root samples from on plant. And how many pieces were incubated for one root sample?

Response and correction of the manuscript based on the comments (see document attached)

Each of the roots was cut into five non-contiguous segments of 1 cm length.

L187: This part is not clear. Are there 4 treatments based on Pb concentration, and 10 pots for each treatment, which is 40 pots together? And 2 seeds in one pot makes the experiment included 80 replicates in 40 pots, is it correct? Then how do you inoculate spore suspension to the seeds? Are the non-inoculated and inoculated seeds in a same pots or in separate pots?

Response and correction of the manuscript based on the comments (see document attached)

In the following experiment, four incremental concentrations of galena (Pb0 = 0, Pb400 = 400, Pb600 = 600, Pb800 = 800 galena mg/kg) were mixed separately with 5 kg of John Innes No. 2 compost (Westland Garden Health, pH 5.5-6.0 at 20˚C, https://www.gardenhealth.com/westland-john-innes-no-2-potting-on-compost) and then, divided between 20 1L pots for a total of 80 pots. Then, 2 seeds of Festuca arundinacea (cultivar Jesup, endophytes-free Nil, type continental, accession number: T9904, % seedling emergence of 92%; supplied by Grasslanz Technology Limited, New Zealand) were sown in each pot at 1 cm depth. The 20 pots for each concentration of galena were equally split between control and endophytic set by inoculating the seeds with either 200 µL of sterile water for the control or 200 µL of the microbial inoculate for the endophytes set, for a total of 10 pots for each treatment set (see Figure 1). The pots were then labelled and randomly distributed in the growth cabinet.

Figure 1. Design of the in vivo experiment. Galena at each concentration was added separately to the soil and well mixed before to dividing into 20 pots for each treatment for a total of 80 pots. In each pot, two seeds of Festuca arundinacea were sown. Pots labelled E were treated with the endophyte consortium and those labelled C were treated with sterile water.

L216: Please mention briefly here how the fresh and dry biomass is measured.

Response:

Before the paragrapher “2.5 Data analysis during the experiment and post-harvest, I briefly described how the preparation of fresh and dry material had been carried ahead.

Reviewer 2 Report

GENERAL

The paper was interesting, important, well written and easy to read.                                   Maintain the past tense (e.g. Line 294).

ABSTRACT

It would be desirable to indicate the relative/quantitative difference observed for those few results where statistically significant differences were observed. 

MATERIALS & METHODS

Line 175 The current name for tall fescue (TF) is Lolium arundinaceum (Schreb.) Darbysh., 1993.  Festuca arundinacea Schreb., 1771 is a homotypic genbank synonym. The seed used is undescribed and it is puzzling why it had to be sourced from New Zealand as there are many turf and pasture cultivars developed and marketed in Ireland, UK and in Europe. Some NZ Grasslands cultivars contain a select a endophyte and some uncertified seedlots contain the wild type endophyte. The type of TF (continental or Mediterranean), the cultivar and the current germination test for the seedlot must be stated. Also, indicate if it was certified seed and what its endophyte status was – presumably you used an endophyte-free seedlot. I wonder why you did not examine local populations of Agrostis from non-contaminated soil?

L189 Was an analysis of the compost supplied? 

L198 Source and quality of water?

L 205 The Germ index was a sensible innovation for rating the various data. Hopefully the seedlot used was fresh and one with a high germ%.

DISCUSSION

The other mechanism for adapting to heavy metal contamination is evolution. In 1950-80, Bangor Univ ecologists, Davies, RW Snaydon and AD Bradshaw reported on the genetic diversity they observed in populations of Agrostis spp. As well as the highly adaptive Anthoxanthum odoratum growing at varying distances from the side of roads bearing heavy traffic. They also studied grasses from various heavy metals from mine sites.

It would be helpful to finish with advice on future work and any limitations of the methods used. Was it appropriate to restrict observations to 50 days? With biomass data, should you assess root length and growth? I am concerned about non-germinating plants.  Would it be better to sow five seeds and thin to two to avoid the risk of sowing a non-viable seed? If only one plant germinated, without competition, its growth could be greater than if two had germinated.  

Author Response

ABSTRACT

POINT 1. It would be desirable to indicate the relative/quantitative difference observed for those few results where statistically significant differences were observed. 

Correction of the manuscript based on the comments (see document attached)

Compared with the control, endophyte treated plants germinated more than 1-day earlier and produced 35.91% more plant tillers at 35 days-after-sowing.

MATERIALS & METHODS

POINT 2. Line 175 The current name for tall fescue (TF) is Lolium arundinaceum (Schreb.) Darbysh., 1993.  Festuca arundinacea Schreb., 1771 is a homotypic genbank synonym.

Correction of the manuscript based on the comments (see document attached)

POINT 3. The seed used is undescribed and it is puzzling why it had to be sourced from New Zealand as there are many turf and pasture cultivars developed and marketed in Ireland, UK and in Europe. Some NZ Grasslands cultivars contain a select a endophyte and some uncertified seedlots contain the wild type endophyte. The type of TF (continental or Mediterranean), the cultivar and the current germination test for the seedlot must be stated. Also, indicate if it was certified seed and what its endophyte status was – presumably you used an endophyte-free seedlot.

Response and correction of the manuscript based on the comments (see document attached)

Festuca arundinacea: cultivar Jesup, endophytes-free Nil, type continental, accession number: T9904, % seedling emergence of 92%; supplied by Grasslanz Technology Limited, New Zealand)

POINT 4. I wonder why you did not examine local populations of Agrostis from non-contaminated soil?

Response:

At the time of the experiment, we were collaborating with the NZ Grasslands company to test a combining effect between their seeds and our endophytic isolates. Indeed, in this experiment, we used an F. arundinacea Nil seed, which was a seed devoid of any microorganisms or Epichloë. This knowledge gave us major support on confirming that the effect observed by the plant were exclusively induced by our endophytes and not by other microorganisms already established inside the seeds. Furthermore, using these commercial seeds gave us a further certainty of the high viability of the seeds, compared to other commercial or natural seeds.

POINT 5. L189 Was an analysis of the compost supplied? 

Response:

No, there wasn’t. However, some further information is available on the official website: https://www.gardenhealth.com/westland-john-innes-no-2-potting-on-compost.

POINT 6. L198 Source and quality of water?

Response:

The water used for the inoculation of the control set was distilled water autoclaved at 121˚C for 15 minutes. Also, the same water was used to prepare the endophytes inoculum.

POINT 7. L 205 The Germ index was a sensible innovation for rating the various data. Hopefully, the seedlot used was fresh and one with a high germ%.

Response:

The seeds used during this experiment were originally from a fresh batch of seeds, immediately delivered to us at the time of the experiment. Moreover, the seedlot were certified with a seedling emergence percentage of 92% and after shipping, they had been stored in a cool and dry place for 6 months before the starting of the experiment. Also, I personally tested the viability of the seed, which confirmed high rate germination closed to 100%.

DISCUSSION

POINT 8. The other mechanism for adapting to heavy metal contamination is evolution. In 1950-80, Bangor Univ ecologists, Davies, RW Snaydon and AD Bradshaw reported on the genetic diversity they observed in populations of Agrostis spp. As well as the highly adaptive Anthoxanthum odoratum growing at varying distances from the side of roads bearing heavy traffic. They also studied grasses from various heavy metals from mine sites.

Correction of the manuscript based on the comments (see document attached)

POINT 9. It would be helpful to finish with advice on future work and any limitations of the methods used.

Response and correction of the manuscript based on the comments (see document attached)

Future research will should focus further on elucidating the mechanisms of heavy metals stress alleviation in the plant and on gaining new insights into the complex relationship between these fungi and their hosts. Thus, isolation of other endo-organisms from different plant tissue could be used to obtain a different variety of microorganisms, which could include, for example, Epichloë and AMF; this would allow us to identify the effect that different classes of microorganisms have in the plant. Indeed, Wężowicz et al. [21] and Vandegrift et al. [22] observed a better performance of the plants when organisms from different classes were combined. This approach will also further reduce the gap between a controlled environment and a natural environment, allowing us to recreate a more realistic natural scenario and microbiota.

POINT 10. Was it appropriate to restrict observations to 50 days?

Response:

It was an arbitrary choice, however, we wanted to collect healthy plants free from disease.

POINT 11. With biomass data, should you assess root length and growth?

Response:

Generally, yes, it would have been useful tested also the root's growth to determinate if the endophytes improved the resistance of the plants in reducing the detrimental effect on the roots. We decided not to analyse the root growth due to the technical difficulties on measuring these appendixes, indeed, collecting roots of a plant grown in soil is not an easy and precise measurement. Moreover, due to the lack of appropriate structure, we avoided to analysis the roots for our health and safety.

POINT 12. I am concerned about non-germinating plants.  Would it be better to sow five seeds and thin to two to avoid the risk of sowing a non-viable seed? 

Response:

In this case, we used commercial seeds with a high germination rate. Furthermore, I personally certified the growth germination rate of the seeds before sowing a couple of months before, which confirmed the high germination rate of the seeds.  

POINT 13. If only one plant germinated, without competition, its growth could be greater than if two had germinated.  

Response:

Yes, indeed, the post-harvest measurements were done comparing the biomass or a total number of tillers produced by each pot and not every single plant. Plants in the same pots at the end were considered as a unique sample, so then the plants were collected together, and their fresh and dry biomass were measured as one single value for each pot.
